# Selection and Reconstruction of Key Locals: A Novel Specific Domain Image-Text Retrieval Method

## ABSTRACT

In recent years, Vision-Language Pre-training (VLP) models have demonstrated rich prior knowledge for multimodal alignment, prompting investigations into their application in Specific Domain Image-Text Retrieval (SDITR) such as Text-Image Person Re-identification (TIReID) and Remote Sensing Image-Text Retrieval (RSITR). Due to the unique data characteristics in specific scenarios, the primary challenge is to leverage discriminative fine-grained local information for improved mapping of images and text into a shared space. Current approaches interact with all multimodal local features for alignment, implicitly focusing on discriminative local information to distinguish data differences, which may bring noise and uncertainty. Furthermore, their VLP feature extractors like CLIP often focus on instance-level representations, potentially reducing the discriminability of fine-grained local features. To alleviate these issues, we propose an **E**xplicit **K**ey **L**ocal information **S**election and **R**econstruction Framework (**EKLSR**), which explicitly selects key local information to enhance feature representation. Specifically, we introduce a **K**ey **L**ocal information **S**election and **F**usion (**KLSF**) that utilizes hidden knowledge from the VLP model to select interpretably and fuse key local information. Secondly, we employ **K**ey **L**ocal segment **R**econstruction (**KLR**) based on multimodal interaction to reconstruct the key local segments of images (text), significantly enriching their discriminative information and enhancing both inter-modal and intra-modal interaction alignment. To demonstrate the effectiveness of our approach, we conducted experiments on five datasets across TIReID and RSITR. Notably, our EKLSR model achieves state-of-the-art performance on two RSITR datasets.

## CCS CONCEPTS

• **Information systems** → **Multimedia and multimodal retrieval**; • **Computing methodologies** → **Neural networks**.

## KEYWORDS

Specifc Domain Image-Text Retrieval, Key Local Information Selection and Reconstruction, Remote Sensing, Text-Image Person Re-identification

*ACM MM, 2024, Melbourne, Australia*
© 2024 Copyright held by the owner/author(s). Publication rights licensed to ACM.
ACM ISBN 978-x-xxxx-xxxx-x/YY/MM
https://doi.org/10.1145/nnnnnnn.nnnnnnn

## 1 INTRODUCTION

With the surge in data proliferation, the Internet and social media platforms have been overwhelmed with an abundance of multimodal content, including both images and text. There is a great demand to automatically retrieve useful information from these vast amounts of data [12, 23, 39, 46, 50, 51]. In response to this demand, significant progress has been made in Cross-modal Image-Text Retrieval (CMITR) in the past few years [12, 17, 23, 46, 52]. Recently, with the emergence of Vision-Language Pre-training (VLP) models [22, 24, 25, 29, 39, 40], CMITR has undergone further advancements.

However, it is worth noting that VLP models like CLIP exhibit less favorable performance in Specific Domain Image-Text Retrieval (SDITR), such as Text-Image Person Re-identification (TIReID) and Remote Sensing Image-Text Retrieval (RSITR), in comparison to their performance in CMITR tasks within the general domain, as depicted in Figure 1(a). The discrepancies in these metrics suggest that specific domains possess unique data characteristics that differ from those of the general domain. Specific domain data typically exhibit high image similarity [53], with semantic nuances often confined to key local segments, such as object regions in images or content-rich words in the text, as seen in Figure 1 (b). Even minor changes in these segments can significantly alter the entire content, highlighting the importance of these segments. Thus, SDITR necessitates that models concentrate on key local segment information [49] to enhance the representation of image-text features in a shared space and improve image-text alignment.

To facilitate the utilization of key local segment information, early bottom-up attention methods [23, 46] captured discriminative local image features, boosting retrieval accuracy in general domains. However, their cross-domain applicability is limited for specific domain image-text tasks. Subsequent advancements have shifted towards leveraging the robust prior knowledge of pre-trained models for specific domains. Instead of directly extracting key local information, these methods [5, 9, 45, 48, 50] employ interactions among all local features in both images and text to implicitly guide the model's attention to key local information. While these methods enhance retrieval performance, they suffer from two main issues.

Firstly, interactions among all local features bring inevitable noise and uncertainty, as they indiscriminately engage with all the local features rather than focusing on key local features. As shown in Figure 1(c), these methods [5, 9, 45, 48, 50] can yield false correlations among extraneous local features, bringing noise and causing misaligned image-text pairs that lower retrieval accuracy [21]. Selecting and utilizing key local features from the multitude is vital for enhancing image-text feature representation. Secondly, VLP models like CLIP [39] have not been optimized for local features, resulting in a lack of discriminability for local features. Throughout the pre-training stage, CLIP predominantly employs contrastive losses that favor instance-level features [49] over local region features.

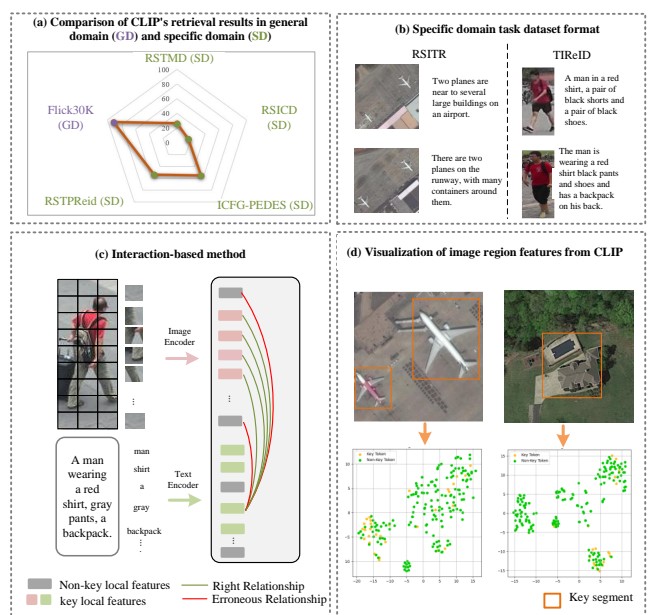

**Figure 1: (a) Fine-tuned CLIP excels on general domain datasets, but its performance drops on four specific domain datasets, revealing limited generalization in specialized domains (R@1 retrieval metric). (b) Highly similar specific domain data. (c) Interaction-based methods can generate noisy associations: "gray" in the text is intended to describe "pants," yet it erroneously associates with the image region of a "gray floor". (d) Key local features are intermingled with other features, lacking distinctiveness.**

Consequently, as shown in Figure 1(d), insufficiently optimized key local features exhibit reduced discriminative capacity and become entangled with other local features. In specific scenarios, the lack of discriminative key local features hampers the differentiation between local segments, thus diminishing the efficacy of subsequent feature interaction modules. Therefore, enhancing the discriminability of key local features is crucial.

To address the aforementioned two issues, we introduce the Explicit Key Local information Selection and Reconstruction (EKLSR) framework based on CLIP. Unlike previous methods [5, 9, 45, 48, 50] that utilize all local features without distinction, our approach not only interpretably selects key local features but also bolsters the discriminability of these features through a multimodal interaction-based reconstruction task. EKLSR consists of a Key Local information Selection and Fusion (KLSF) module and Key Local segment Reconstruction (KLR) based on multimodal interaction.

KLSF directly selects and utilizes key local features from the multitude to enhance image-text feature representation. It leverages hidden knowledge from the CLIP to interpretably select key local information, thereby enhancing the final image-text feature representation. Specifically, it employs VLP model hidden priors to assign an importance factor to each local feature, gauging its significance. In the key feature selection process, local features with higher importance factors are more likely to be selected. Selected key local

features capture object details but lack associations between the objects. Thus, KLSF fuses key local features with instance-level global features representing object relationships, significantly enhancing the final multimodal feature representation. Additionally, KLSF operates independently on image and text branches, maintaining the dual-stream structure of the model and enabling offline computation during inference.

KLR enhances the discriminability of the key local features. Inspired by works [2, 61] that enhance feature specificity through feature reconstruction tasks, KLR applies both Masked Language Modeling (MLM) and Masked Visual Modeling (MVM) tasks to the reconstruction of selected key local features. These two tasks ensure that the features retain their unique information, thereby preserving their discriminative properties. MLM and MVM are generally utilized in the pre-training stage of Visual-Language Pre-training (VLP) [6, 25, 32, 43], we make the first attempt to demonstrate the effectiveness of MLM and MVM in downstream fine-tuning tasks. Innovatively, to ensure the key local segments are reconstructed, we reconstruct key segments selectively rather than random reconstruction of all segments. Furthermore, we predict masked segments through the integration of intra-modal and inter-modal interaction information. This not only strengthens multimodal fine-grained alignment but also aids the backbone network in extracting more discriminative features.

The contributions of this paper can be summarized as follows:

- We propose EKLSR to make CLIP more adaptable to fine-grained downstream tasks without substantial additional supervision and inference costs.
- We propose a Key Local information Selection and Fusion (KLSF) module that selects interpretably key local features from the multitude to enhance image-text feature representation.
- We identify the limited local feature representation capability of CLIP. Therefore, we introduce Key Local segment Reconstruction (KLR) based on multimodal interaction, which strengthens the discriminability of CLIP local features and facilitates inter-modal feature interaction.
- Extensive experiments has been conducted on specific domain image-text retrieval tasks, such as RSITR and TIReID. Notably, our EKLSR model achieves state-of-the-art performance on two benchmark RSITR datasets.

## 2 RELATED WORK

### 2.1 Vision-Language Pre-Training

Vision-Language Pre-training (VLP) aims to learn the semantic correspondence between the vision and text by pre-training on a large-scale dataset. Inspired by the success of Transformer-based language model BERT [8] and vision model ViT [11], most current multimodal models [6, 18, 22, 24, 25, 29, 37, 39, 40, 44] adopt their variants to learn multimodal representations. Existing VLP models can be categorized into two types: single-stream and dual-stream models. Single-stream models [18, 22, 25, 37, 44] combine extracted text and vision local embeddings as input to a Transformer structure to extract a joint representation of image-text pairs. They are more effective in capturing fine-grained relationships between images and text, leading to higher retrieval accuracy

compared to dual-stream models. Furthermore, the complex interaction mechanism leads to slower inference speed, which does not meet the real-time requirements of specific domain tasks. On the other hand, dual-stream models [12, 39, 44] utilize independent encoders to map images or text into global embeddings and align them on the common space. Recent Transformer-based dual-stream models [19, 25, 39] have chosen to improve their performance by leveraging additional large-scale data. Despite their impressive performance in image-text retrieval tasks, their heavy reliance on instance-level representations limits their ability to capture local features effectively. Consequently, they face constraints when applied to Specific Domain Image-Text Retrieval (SDITR) tasks.

## 2.2 Specific Domain Image-Text Retrieval (SDITR)

**TIReID**: TIReID is a multimodal task [38, 56], first introduced by [28]. Compared to general cross-modal retrieval tasks, TIReID demands models to pay more attention to fine-grained information and the correspondence between modalities to distinguish the differences among pedestrians. Initial global matching approaches [57, 58] aligned images and text in a joint embedding space through cross-modal matching loss functions, neglecting direct emphasis on fine-grained local information. Recently, several approaches [5, 9, 45, 48] have emerged that leverage single-modal pre-trained models (such as ResNet [16], ViT [11], and BERT [8]), or VLP models like CLIP, as backbones to incorporate powerful external knowledge. Han et al. [15] first introduced a CLIP model for text-to-image person retrieval. Later, CFine [49] builds upon VLP and introduces a token selection module to directly choose informative key local embeddings. However, this selection process lacks interpretability. IRRA [21] learns relations between local visual-textual tokens and enhances global image-text matching. However, it overlooked the fact that CLIP heavily relies on instance-level representations, resulting in limited capability to represent local features. Our model introduces a novel task of reconstructing key local segment features in images (text) through multimodal interaction, which greatly enhances their discriminative information.

**RSITR**: RSITR refers to recalling required RS images with text. Initial studies [1, 34] prioritized global image-text representations, where multimodal information was encoded and merged to formulate a shared semantic representation. Subsequent research [1, 34, 35, 54] introduced additional information to enhance modality representations. Yuan et al. [54] employed a shared modality transmission module to facilitate communication across modalities. Cheng et al. [7] devised a semantic alignment module to effectively identify latent correspondences between images and text. Yuan et al. [53] developed an asymmetric multimodal feature matching network and employed multi-scale feature information. Later, [30, 31, 55] found that the knowledge CLIP can be transferred to the remote sensing domain, but there is a lack of in-depth research on remote sensing image-text characteristics. In this study, we employed CLIP as the backbone and introduced the KLSF module to extract fused discriminative local features, mitigating the issue of high similarity of remote sensing image-text data.

## 3 METHOD

In this section, we will introduce our proposed Explicit Key Local information Selection and Reconstruction (EKLSR) framework. An overview of EKLSR is shown in Figure 2 (a). It consists of a dual-stream feature extraction backbone, Key Local information Selection and Fusion (KLSF), and Key Local segment Reconstruction (KLR) based on multimodal interaction. We leverage CLIP [39] as the initialization of the backbone. KLSF leverages hidden knowledge of CLIP to select interpretably and fuse key local features to form the final feature representations of the image and text. KLR masks key segments of images and text, and it predicts these segments using the contextual information from unmasked image (text) and the global information from the paired text (image). KLR does not participate in the inference process. The details of EKLSR will be discussed in the following subsections.

## 3.1 Feature Extraction Backbone

Inspired by the success of transferring knowledge from CLIP [39] to text-image retrieval [21], we directly initialize our EKLSR framework with the CLIP pre-trained model weight.

**Image Encoder.** Given an input image $I \in R^{H \times W \times C}$, we employ the CLIP pre-trained Vision Transformer model to obtain both global and local token embeddings of the image. Firstly, the image I is split into a sequence of $N = W \times H/P^2$ fixed-size patches, where P represents the patch size. The patch sequence is then mapped to a token sequence $\{v_i\}_{i=1}^{N}$ through a trainable linear projection. By injecting position embeddings and an additional [CLS] token, the token sequence $V = \{v_{cls}, v_1, \ldots, v_N\}$ is inputted to the Transformer, enabling multi-head self-attention to obtain $\left\{f_{cls}^v, f_1^v, f_2^v, \ldots, f_N^v\right\}$. Finally, the global text feature $f_g^v$ is obtained by linearly mapping $f_{cls}^v$.

**Text Encoder.** For an original text $T = \{t_1, t_2, \ldots, t_n\}$ where $t_i$ denotes the i-th word in the text, consisting of n words. We first tokenize the input text T and add [SOS] and [EOS] markers at the beginning and end respectively, forming $T = \{t_{sos}, t_1, \ldots, t_n, t_{eos}\}$. It is then fed into text Transformer, where self-attention is employed to learn global dependencies, obtaining $\left\{f_{sos}^t, f_1^t, f_2^t, \ldots, f_n^t, f_{eos}^t\right\}$. The global text feature $f_g^t$ is obtained by linearly mapping $f_{sos}^t$.

## 3.2 Key Local Information Selection and Fusion (KLSF)

*3.2.1 Interpretable Key Local Feature Selection.* To enhance the application of CLIP in specific domain retrieval tasks, it is critical to concentrate on extracting key local segment information, such as emphasizing subject origins rather than the background in images and focusing on content-rich words (pronouns, verbs, adjectives, adverbs, nouns) instead of function words (prepositions, conjunctions, etc.) in text. Previous interaction-based methods that utilize all local features tend to bring noise, and relying on interactions to highlight key local information can lead to uninterpretability and uncertainty. To mitigate these issues, we propose an interpretable method for assessing the importance of local features and selecting key local features. This process begins by leveraging the strong image-text understanding capabilities of CLIP to calculate

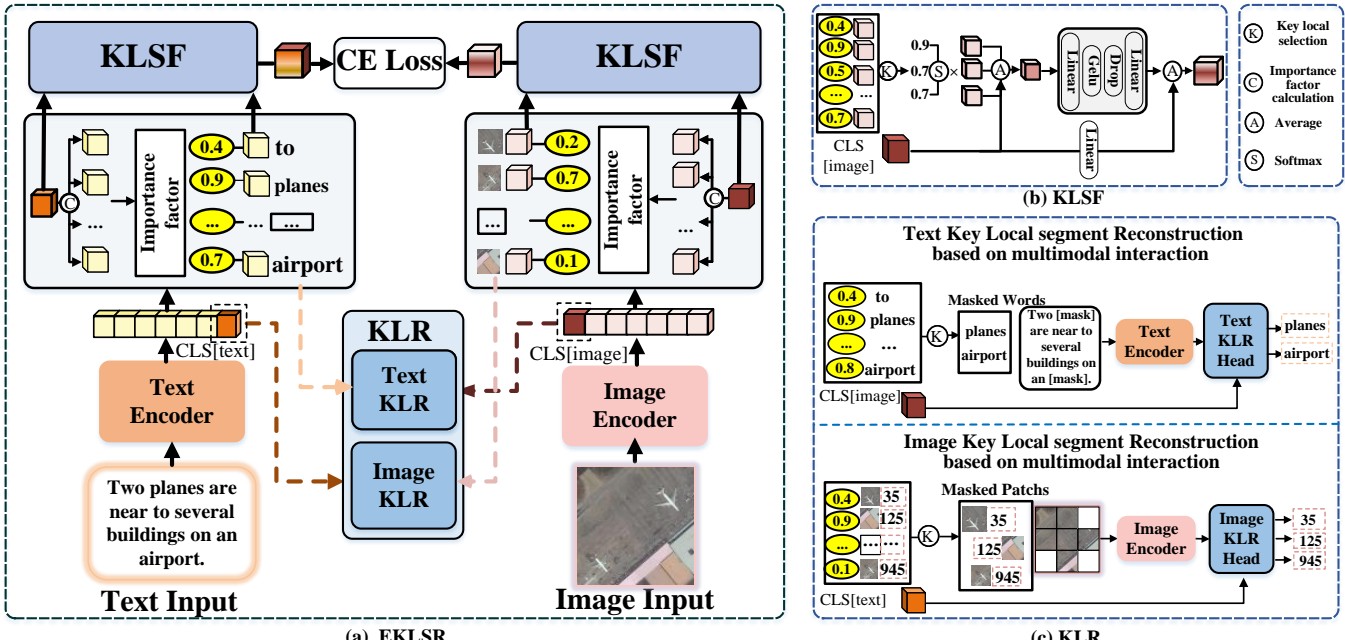

Figure 2: (a) Explicit Key Local Information Selection and Reconstruction (EKLSR) framework for image-text retrieval. (b) Key Local information Selection And Fusion (KLSF) module. (c) Key Local segment Reconstruction based on multimodal interaction (KLR) tasks.

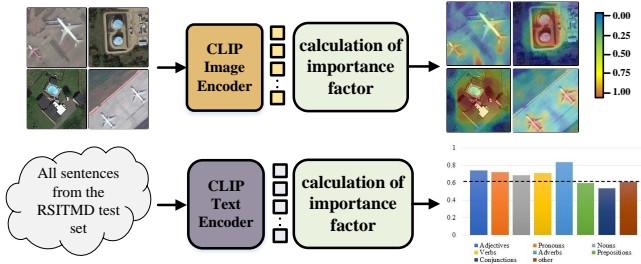

Figure 3: Importance factor distribution. It reveals that regions with high importance factors in images correspond to the primary subject regions. In the text, the average importance factors of content-rich words (pronouns, verbs, adjectives, adverbs, nouns) are higher than those of function words (prepositions, conjunctions, etc.).

the importance factor of each local feature, as follows:

$$s_i^v = 1 - \text{cosine}\left(f_{cls}^v, f_i^v\right) \quad (1)$$

$$s_i^t = 1 - \text{cosine}\left(f_{cls}^t, f_i^t\right) \quad (2)$$

Where $f_{cls}^v$ and $f_{cls}^t$ denote instance-level features of images and text, respectively, while $f_i^v$ and $f_i^t$ represent their local features. $s_i^v(s_i^t)$ stands for the importance factor of i-th local feature in images (text). We visualized the distribution of image and text importance factors on the test set of the RSTMD dataset, as shown in Fig. 3. The visualization indicates that the segments of high importance factors

correlate with the main subject regions in images and meaningful words in the text, confirming that these factors can adaptively gauge the importance of local features. Furthermore, we interpretably select key local features based on their importance factors. We employ a polynomial probability selection method that selects features based on their importance factors. Specifically, the likelihood of being selected increases proportionally with the magnitude of their importance factor. This method can select not only the majority of key local features but also a few non-key local features that are beneficial for feature representation, outperforming random and top-k selection methods. Through polynomial probability selection, key local features are selectively extracted.

$$ID_v = \left\{id_i^v\right\}_{i=1}^{\alpha*N} = \text{PolynomialSelection}\ (S_v, \alpha) \quad (3)$$

where $S_v = \{s_i\}_{i=0}^N$, $\alpha$ represents the proportion of key local feature selection in the image, and the number of selected key features is $\alpha * N$. $ID_v = \left\{id_i^v\right\}_{i=1}^{\alpha*N}$ indicates the id set of selected key image local features. The selection method for key features of text is similar.

$$ID_t = \left\{id_i^t\right\}_{i=1}^{\beta*n} = \text{PolynomialSelection}\ (S_t, \beta) \quad (4)$$

where $\beta$ represents the proportion of key segment embedding selection in the text. $ID_t = \left\{id_i^t\right\}_{i=1}^{\beta*n}$ indicates the id set of selected key text local features.

*3.2.2 Fusion of Global and Key Local Features.* Key local features selected capture local details but lack associations among these details. Therefore, we fuse key local tokens with instance-level global features that represent relationships among local details to

foster complementarity between their respective feature information. As shown in Figure 2 (b), for images, we apply Softmax to the importance factors to serve as weight coefficients for the key local features. We multiply these coefficients by the key local features and accumulate the results to obtain the key fused representation. To prevent the loss of global contextual features, we integrate it with the key fused representation using a residual connection, resulting in the joint feature representation $f_{\text{key}}^v$. The procedure is shown below:

$$f_{\text{key}}^v = \sum_{k \in ID_v} \frac{\exp(s_k)}{\sum_{i=1}^{N} \exp(s_i)} \cdot f_k^v + f_{cls}^v \tag{5}$$

The joint feature $f_{\text{key}}^v$ may still contain redundant feature representations or even misleading information [54]. Naturally, we perform a secondary feature transformation on the $f_{\text{key}}^v$ to suppress irrelevant information. We map the joint feature to a high-dimensional hidden space, randomly deactivating some neurons to filter out redundant and irrelevant information. Subsequently, we linearly map the result to the common space. These steps are formulated as follows:

$$f_{\text{keyh}}^v = \text{Dropout}\left(\text{GELU}\left(\text{Linear}\left(f_{\text{key}}^v\right)\right)\right) \tag{6}$$

$$\hat{f}_{\text{key}}^v = \text{Linear}\left(f_{\text{keyh}}^v\right) \tag{7}$$

The global representation $f_g^v$ extracted from CLIP contains valuable hidden knowledge of the global relationship, which is crucial in the final feature representation. The fusion is performed as follows:

$$F^v = \hat{f}_{\text{key}}^v + f_g^v \tag{8}$$

where $F^v$ is the final image feature. A similar process is performed for the text modality to obtain the final text feature $F^t$.

Finally, we use contrastive loss for optimization. The loss function can be expressed as

$$L_{ce} = -\frac{1}{2m} \sum_{j=1}^{B} \log \frac{\exp\left(\text{cosine}\left(F_j^v, F_j^t\right)\right)}{\sum_{k=1}^{B} \exp\left(\text{cosine}\left(F_j^v, F_k^t\right)\right)} - \frac{1}{2m} \sum_{j=1}^{B} \log \frac{\exp\left(\text{cosine}\left(F_j^t, F_j^v\right)\right)}{\sum_{k=1}^{B} \exp\left(\text{cosine}\left(F_j^t, F_k^v\right)\right)} \tag{9}$$

where $F_j^v$ is the final feature representation of the j-th image in the current batch data. This is a bidirectional loss function, where the first half calculates the loss for the image-to-text retrieval task, and the second half represents the loss calculation for the text-to-image retrieval task. This loss function aims to bring paired image-text features closer together in a common space while pushing unrelated pairs further apart.

## 3.3 Key Local Segment Reconstruction based on Multimodal Interaction (KLR)

CLIP has not been optimized for local feature representation, resulting in a lack of discriminability for local features. This does not facilitate the mutual complementation of local feature information. We propose a KLR task to augment the discriminability of local features. Unlike previous methods [18, 22, 25, 37, 44] that randomly reconstruct segments, KLR ensures the reconstruction of key local segments and enhance the interaction between multimodal local features. The KLR is only utilized during model training and is not employed during inference.

*3.3.1 Text Key Local Segment Reconstruction based on Multimodal Interaction (Text KLR).* We employ an approach similar to MLM of BERT [8] to reconstruct the key segments. However, there are two points of difference as shown in the upper part of Figure 2(c). Firstly, to ensure the masking of key local tokens, we select segments with higher importance factors in a proportion of $\gamma$. Then, we apply masking to the selected segments. Specifically

$$\widehat{T} = \text{Mask}\left(T, \text{PolynomialSelection}\left(S_t, \gamma\right)\right) \tag{10}$$

Where $\widehat{T} = \left\{t_{\text{sos}}, t_1, t_2^{\text{mask}}, t_3^{\text{mask}} \ldots, t_n, t_{\text{eos}}\right\}$ denotes the text encoding after masking. In this case, the number of masked words is $\gamma * n$. Subsequently, $\widehat{T}$ is input to the text Transformer encoder:

$$\widehat{F}_t = \text{Transformer}(\widehat{T}) \tag{11}$$

where $\widehat{F}_t = \left\{\hat{f}_{\text{sos}}^t, \hat{f}_1^t, \hat{f}_2^{\text{tmask}}, \hat{f}_3^{\text{tmask}}, \ldots, \hat{f}_n^t, \hat{f}_{\text{eos}}^t\right\}$ is masked text features.

Secondly, to enhance the interaction between image and text modalities, we predict masked text segments with unmasked text segment features and paired image instance-level features. The prediction process can be expressed as:

$$\hat{y}_i^t = \text{Linear}\left(\hat{f}_i^{\text{tmask}} + f_g^v\right) \tag{12}$$

$$L_{tklr} = -\frac{1}{B} \sum_{j=1}^{B} \sum_{i=1}^{\beta * n} \left(y_{i,j}^t \log \hat{y}_{i,j}^t + \left(1 - y_{i,j}^t\right) \log \left(1 - \hat{y}_{i,j}^t\right)\right) \tag{13}$$

Where $y_{i,j}$ denotes the label at the j-th mask position of the i-th text. $\hat{y}_{i,j}^t$ denotes the predicted result.

*3.3.2 Image Key Local Segment Reconstruction based on multimodal interaction (Image KLR).* To reconstruct the key local segment features in the image, we adopt a similar approach to Text KLR. As shown in the lower part of Figure 2 (c).

$$\widehat{V} = \text{Mask}\left(V, \text{PolynomialSelection}\left(S_v, \tau\right)\right) \tag{14}$$

$\tau$ represents the probability of masking, and the number of masked image segments is $\tau * N$. The masked image $\widehat{V}$ is then input to the image encoder, obtaining $\hat{F}_v = \left\{f_{cls}^v, f_1^v, f_2^{\text{vmask}}, f_3^v, f_4^{\text{vmask}}, \ldots, f_N^v\right\}$.

$$\widehat{F}_v = \text{Transformer}(\widehat{V}) \tag{15}$$

We tokenize image segments $\{v_i\}_{i=1}^{N}$ using the image tokenizer of BEIT [3], obtaining labels $Y_v = \left\{y_1^v, y_2^v, \ldots, y_{N*\alpha}^v\right\}$ for all masked patches. Then, we predict the labels of masked image segments with the unmasked image segment features and paired text global features.

$$\hat{y}_i^v = \text{Linear}\left(\hat{f}_i^{\text{vmask}} + f_g^t\right) \tag{16}$$

$$\mathcal{L}_{iklr} = -\frac{1}{B} \sum_{i=1}^{B} \sum_{j=1}^{\beta * n} \left(y_{i,j}^v \log \hat{y}_{i,j}^v + \left(1 - y_{i,j}^v\right) \log \left(1 - \hat{y}_{i,j}^v\right)\right) \tag{17}$$

Finally, the total loss of our EKLSR framework is defined as:

$$\mathcal{L} = \mathcal{L}_{ce} + \lambda \cdot \mathcal{L}_{tklr} + \eta \cdot \mathcal{L}_{iklr} \tag{18}$$

where $\mathcal{L}_{ce}$ denotes the image-text matching loss. $\mathcal{L}_{tklr}$ and $\mathcal{L}_{iklr}$ respectively denote the reconstruction loss of key segments for the text and image.

## 4 EXPERIMENTAL

### 4.1 Datasets

In this paper, we experimented with two Remote Sensing Image-Text Retrieval (RSITR) datasets and three Text-Image Person Re-identification (TIReID) datasets.

*4.1.1 RSITR datasets.* We conducted experiments on two remote sensing image-text datasets, RSICD [33], RSITMD [53]. The RSICD dataset consists of 10,921 RS images in 30 semantic categories, each of which has 5 captions. The RSITMD dataset consists of 4743 images in 32 semantic categories, with a total of 23,715 captions.

*4.1.2 TIReID datasets.* We conducted experiments on three TIReID datasets, CUHK-PEDES [27], ICFG-PEDES [10], RSTPReid [60]. The CUHK-PEDES contains 40,206 images and 80,412 textual descriptions for 13,003 identities. The ICFG-PEDES contains a total of 54,522 images for 4,102 identities. Each image has only one corresponding textual description. RSTPReid contains 20,505 images of 4,101 identities from 15 cameras.

### 4.2 Metrics and Implementation Details

*4.2.1 Evaluation Metrics.* The experimental metrics are R@K and mR, where R@K (K=1, 5, and 10) is defined as the similarity ranking of matching pairs included in the top K retrieval results, and a higher value of R@K indicates better performance. mR represents the average of all R@K, which is more reasonable for evaluating the overall performance of the model.

*4.2.2 Implementation Details.* For KLSF, the probability of selecting key features in the image, $\alpha$, is set to 0.3 for RSITR and 0.55 for TIReID. The probability of selecting key features in the text, $\beta$, is set to 0.3 for RSITR and 0.70 for TIReID. For KLR, the probability of masking the image, $\tau$, is set to 0.5 for RSITR and 0.2 for TIReID, and the probability of masking the text, $\gamma$, is set to 0.5 for RSITR and 0.2 for TIReID. Both $\lambda$ and $\eta$ are set to 0.1. We employ a cosine annealing learning rate strategy with a warm-up period of 2000 steps. The learning rate is set to 1e-05, the batch size is 128, and the total number of epochs is 10 for RSITR and 50 for TIReID. The initialization of our framework is based on CLIP-ViT-B/16. The experiments are conducted using the PyTorch on an NVIDIA RTX 3090 GPU for RSITR task and an A100 GPU for TIReID task.

### 4.3 Comparison with State-of-the-Art Methods

We conducted an extensive evaluation of the EKLSR performance across two specific domains: RSITR (RSITMD and RSICD) and TIReID (CUHK-PEDE, ICFG-PEDES, and RSTPReid). We benchmarked our EKLSR against the best comparison model and the CLIP (baseline), as depicted in Figure 4, clearly demonstrating our method's superior performance. Detailed analyses for each dataset are presented below.

*4.3.1 RSITR Results.* In this section, we compare our approach with state-of-the-art methods on two RSITR benchmark datasets,

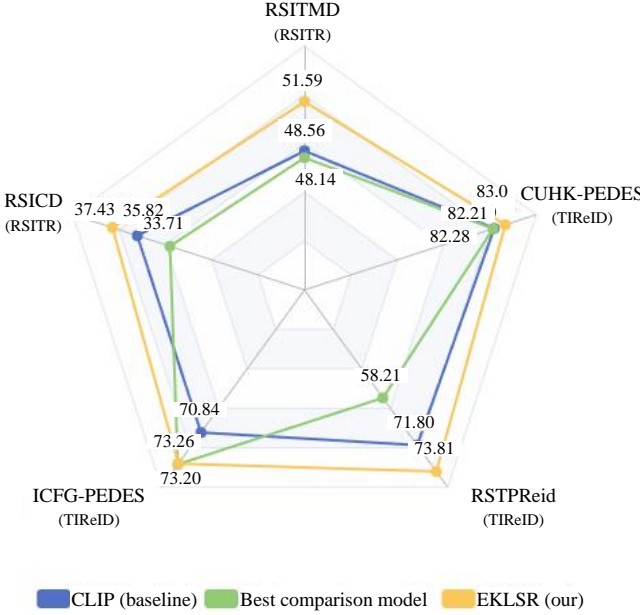

**Figure 4: Comparison of EKLSR with other models based on the mR metric across two RSITR (RSITMD, RSICD) and three TIReID (CUHK-PEDE, ICFG-PEDES, RSTPReid) datasets.**

RSICD and RSITMD, as shown in Table 1. The "Type" column classifications "F," "R," "V," and "C" correspond to image backbones based on bottom-up attention models, ResNet, VIT, and CLIP, respectively. Text Retrieval refers to image-to-text retrieval. Image Retrieval refers to text-to-image retrieval.

**Performance Comparisons on RSITMD.** As indicated in Table 1, EKLSR outperforms all methods across all recall rates R@K, achieving mR accuracy of 51.59%, which is 3.03% higher than the baseline and 3.45% higher than the current best method, TGKT [30]. Notably, our directly fine-tuned CLIP baseline already surpasses the advanced TGKT [30] method with an mR accuracy of 48.56%. It is evident from the "Type" column in Table 1 that the robust feature extraction backbones in RSITR are key, with VLP-based methods becoming increasingly dominant. This underscores the significance of our research in optimizing VLP models for specific domain tasks.

**Performance Comparisons on RSICD.** As shown in Table 1, the baseline exceeds the most recent state-of-the-art results by +2.11% in mR accuracy. Furthermore, our proposed EKLSR surpasses all methods across all R@K accuracy, significantly outperforming the latest advanced method, TGKT, by +3.73% in mR accuracy.

EKLSR consistently delivers state-of-the-art performance across all metrics on two RSITR datasets. This underscores the efficacy of EKLSR in effectively leveraging VLP model knowledge for RSITR.

*4.3.2 TIReID Results.* In this section, we evaluate the generalization capability of our proposed EKLSR model on TIReID by conducting experiments across three public TIReID benchmark datasets (CUHK-PEDES, ICFG-PEDES, and RSTPReid).

**Performance Comparisons on CUHK-PEDES.** We evaluated our EKLSR model on the widely-used CUHK-PEDES dataset, with performance comparisons shown in Table 2. EKLSR achieved an

| | | RSITMD | | | | | | | RSICD | | | | | | |
| | | Text Retrieval | | | Image Retrieval | | | | Text Retrieval | | | Image Retrieval | | | |
| | Type | R@1 | R@5 | R@10 | R@1 | R@5 | R@10 | mR | R@1 | R@5 | R@10 | R@1 | R@5 | R@10 | mR |
|---|---|---|---|---|---|---|---|---|---|---|---|---|---|---|---|
| SCAN [23] ECCV'18 | F | 11.06 | 25.88 | 39.38 | 9.82 | 29.38 | 42.12 | 26.28 | 5.85 | 12.89 | 19.84 | 3.71 | 16.40 | 26.73 | 14.23 |
| CAMP [46] ICCV'19 | F | 11.73 | 26.99 | 38.05 | 8.27 | 27.79 | 44.34 | 26.20 | 5.12 | 12.89 | 21.12 | 4.15 | 15.23 | 27.81 | 14.39 |
| AMFMN [53] TGRS'22 | R | 11.06 | 29.20 | 38.72 | 9.96 | 34.03 | 52.96 | 29.32 | 5.39 | 15.08 | 23.40 | 4.90 | 18.28 | 31.44 | 16.42 |
| GaLR [54] TGRS'22 | R | 14.82 | 31.64 | 42.48 | 11.15 | 36.68 | 51.68 | 31.41 | 6.59 | 19.85 | 31.04 | 4.69 | 19.48 | 32.13 | 18.96 |
| SWAN [36] ICMR'23 | R | 13.35 | 32.15 | 46.90 | 11.24 | 40.40 | 60.60 | 34.11 | 7.41 | 20.13 | 30.86 | 5.56 | 22.26 | 37.41 | 20.61 |
| KAMCL [20] TGRS'23 | R | 16.51 | 36.28 | 49.12 | 13.50 | 42.15 | 59.32 | 36.14 | 12.08 | 27.26 | 38.70 | 8.65 | 27.43 | 42.51 | 26.10 |
| MSITA [4] TGRS'24 | V | 15.22 | 34.2 | 47.65 | 12.15 | 39.92 | 57.72 | 34.48 | 8.67 | 22.71 | 33.91 | 6.13 | 21.98 | 35.39 | 21.47 |
| VIT+BERT | V | 12.83 | 31.19 | 46.24 | 9.60 | 36.59 | 54.42 | 31.81 | 9.06 | 22.78 | 32.75 | 5.32 | 19.47 | 33.71 | 20.52 |
| ResNet101+ BERT | R | 13.50 | 32.30 | 46.24 | 11.90 | 36.46 | 52.43 | 32.14 | 9.15 | 23.70 | 35.32 | 5.07 | 19.69 | 33.21 | 21.02 |
| RemoteCLIP [31] arXiv'23 | C | 22.79 | 49.12 | 61.50 | 18.14 | 51.73 | 70.09 | 45.89 | 15.83 | 36.51 | 51.69 | 12.42 | 34.38 | 51.27 | 33.68 |
| TGKT [30] IGRS'24 | C | 25.88 | 50.00 | 63.05 | 20.58 | 55.18 | 74.16 | 48.14 | 15.37 | 35.50 | 50.50 | 12.83 | 36.21 | 51.88 | 33.71 |
| Baseline (CLIP-ViT-B/16) | C | 25.88 | 50.22 | 63.27 | 23.14 | 56.11 | 72.74 | 48.56 | 19.21 | 38.15 | 50.59 | 14.07 | 38.50 | 54.40 | 35.82 |
| EKLSR(ours) | C | **30.08** | **53.76** | **66.15** | **27.87** | **57.61** | **74.11** | **51.59** | **19.48** | **39.98** | **53.33** | **15.33** | **39.92** | **56.54** | **37.43** |

**Table 1: Experimental results on RSITR task.**

| Method | Type | R@1 | R@5 | R@10 | mR |
|---|---|---|---|---|---|
| TIMAM [41] ICCV'19 | R | 54.51 | 77.56 | 79.27 | 70.44 |
| ViTAA [45] ECCV'20 | R | 54.92 | 75.18 | 82.9 | 71.00 |
| NAFS [13] arXiv'21 | R | 59.36 | 79.13 | 86.00 | 74.83 |
| LBUL [47] MM'22 | R | 64.04 | 82.66 | 87.22 | 77.97 |
| TIPCB [5] Neuro'22 | R | 64.26 | 83.19 | 89.10 | 78.85 |
| SAF [26] ICASSP'22 | R | 64.13 | 82.62 | 88.40 | 78.38 |
| IVT [42] ECCVW'22 | R | 65.59 | 83.11 | 89.21 | 79.30 |
| CFine [49] arXiv'22 | C | 69.57 | 85.93 | 91.15 | 82.21 |
| TGDA [14] TCSVT'23 | R | 64.64 | 83.38 | 89.34 | 79.12 |
| Baseline (CLIP-ViT-B/16) | C | 67.91 | 86.98 | 91.95 | 82.28 |
| EKLSR (ours) | C | **69.62** | **87.34** | **92.05** | **83.00** |

**Table 2: Experimental Evaluation on CUHK-PEDES for TIReID.**

| Method | Type | R@1 | R@5 | R@10 | mR |
|---|---|---|---|---|---|
| DSSL [59] MM'21 | R | 39.05 | 62.60 | 73.95 | 58.53 |
| SSAN [9] arXiv'21 | R | 43.50 | 67.80 | 77.15 | 62.81 |
| LBUL [47] MM'22 | R | 45.55 | 68.20 | 77.85 | 63.86 |
| IVT [42] ECCVW'22 | V | 46.70 | 70.00 | 78.80 | 65.16 |
| CFine [49] arXiv'22 | C | 50.55 | 72.50 | 81.60 | 68.21 |
| TGDA [14] TCSVT'23 | R | 48.35 | 73.15 | 80.30 | 67.26 |
| Baseline (CLIP-ViT-B/16) | C | 53.60 | 77.20 | 84.60 | 71.80 |
| EKLSR (ours) | C | **55.30** | **79.15** | **87.00** | **73.81** |

**Table 3: Experimental Evaluation on RSTPReid for TIReID.**

| Method | Type | R@1 | R@5 | R@10 | mR |
|---|---|---|---|---|---|
| CMPM/C [57] ECCV'18 | R | 43.51 | 65.44 | 74.26 | 61.07 |
| ViTAA [45] ECCV'20 | R | 50.98 | 68.79 | 75.78 | 65.18 |
| SSAN [9] arXiv'21 | R | 54.23 | 72.63 | 79.53 | 68.79 |
| IVT [42] ECCVW'22 | V | 56.04 | 73.60 | 80.22 | 69.95 |
| CFine [49] arXiv'22 | C | 60.83 | 76.55 | 82.42 | 73.26 |
| TGDA [14] TCSVT'23 | R | 57.26 | 75.19 | 81.80 | 71.41 |
| Baseline (CLIP-ViT-B/16) | C | 55.23 | 75.38 | 81.90 | 70.84 |
| EKLSR (ours) | C | **59.03** | **77.26** | **83.57** | **73.29** |

**Table 4: Experimental Evaluation on ICFG-PEDES for TIReID.**

R@1 accuracy of 51.59%, surpassing the baseline by 1.71%. Moreover, when compared to the strongest competitor, a similar CLIP-based method CFine [49], EKLSR reached 87.34% (+1.41%) and 92.05% (+0.9%) in Rank-5 and Rank-10 accuracy, respectively. These results validate the effectiveness of our proposed key local selection and enhancement strategies (KLSF and KLR) in bridging the modality gap critical for TIReID tasks.

**Performance Comparisons on ICFG-PEDES and RSTPReid.** We compared our EKLSR against previous works on two additional benchmarks, RSTPReid and ICFG-PEDES, as illustrated in Tables 3 and Table 4. EKLSR consistently outperforms the baselines on both datasets. Specifically, on the RSTPReeid dataset, EKLSR significantly surpasses the same CLIP-based method, CFine, achieving 55.30% (+4.95) and 73.81% (+5.60%) in Rank-1 and mR accuracy, respectively. Similarly, on the ICFG-PEDES, it significantly outperforms the baseline and achieves performance comparable to the CFine. These results demonstrate the robustness and generalizability of EKLSR.

## 4.4 Ablation Experiments

To fully demonstrate the impact of different components in EKLSR, we use the CLIP-ViT-B/16 models as the baseline and conduct experiments on the RSITMD (RSITR task) and CUHK-PEDES (TIReID task) datasets. Refer to Table 5 for the experimental results. The "ours (w/o interaction)" denotes that the global information from paired modalities is not used during masked segment reconstruction.

(1) By comparing No.0 and No.1, we demonstrate the effectiveness of the KLSF module, showing an approximately 1% improvement in mR accuracy on the RSITMD and CUHK-PEDES datasets. This indicates that the KLSF module can extract key local information from the powerful multimodal knowledge representation of CLIP to enhance the final feature representation.

(2) Comparing No.2 and No.1 on the RSITMD dataset shows text retrieval gains with R@1, R@5, and R@10 increasing by 0.44%, 1.77%, and 1.11%, respectively. However, image retrieval improvements are negligible across datasets. This can be attributed to Text KLR's focus on text reconstruction, only improving the discriminability of text local features and benefiting unidirectional text

| No. | Methods | Components | | | RSITMD | | | | | | | CUHK-PEDES | | | |
|---|---|---|---|---|---|---|---|---|---|---|---|---|---|---|---|
| | | KLSF | Text KLR | Image KLR | Text Retrieval | | | Image Retrieval | | | mR | Image Retrieval | | | mR |
| | | | | | R@1 | R@5 | R@10 | R@1 | R@5 | R@10 | | R@1 | R@5 | R@10 | |
| 0 | baseline | | | | 25.88 | 50.22 | 63.27 | 68.66 | 56.11 | 72.74 | 48.56 | 67.91 | 86.98 | 91.95 | 82.28 |
| 1 | +KLSF | ✓ | | | 29.86 | 50.66 | 64.15 | 23.27 | 56.90 | 73.14 | 49.66 | 68.66 | 86.82 | 91.84 | 82.44 |
| 2 | +KTSF+Text KLR | ✓ | ✓ | | 30.30 | 52.43 | 65.26 | 23.31 | 55.92 | 72.61 | 49.97 | 69.16 | 87.11 | 92.17 | 82.81 |
| 3 | +KTSF+Image KLR | ✓ | | ✓ | 28.53 | 52.65 | 65.26 | 25.00 | 57.56 | 73.45 | 50.40 | 69.28 | 86.85 | 91.81 | 82.65 |
| 4 | ours (w/o interaction) | ✓ | ✓ | ✓ | 28.70 | 53.57 | 65.5 | 27.21 | 56.36 | 73.59 | 50.82 | 69.16 | 87.11 | 92.17 | 82.81 |
| 5 | ours | ✓ | ✓ | ✓ | 30.08 | 53.76 | 66.15 | 27.87 | 57.61 | 74.11 | 51.59 | 69.62 | 87.34 | 92.05 | 83.00 |

**Table 5: Ablation study on each component of EKLSR on RSITMD and CUHK-PEDES.**

| No. | Module | Selection Method | | | RSITMD | | | | | | | CUHK-PEDES | | | |
|---|---|---|---|---|---|---|---|---|---|---|---|---|---|---|---|
| | | random | topk | our | Text Retrieval | | | Image Retrieval | | | mR | Image Retrieval | | | mR |
| | | | | | R@1 | R@5 | R@10 | R@1 | R@5 | R@10 | | R@1 | R@5 | R@10 | |
| 0 | | ✓ | | | 25.22 | 46.46 | 61.73 | 22.35 | 55.40 | 73.14 | 47.38 | 64.71 | 84.81 | 90.62 | 80.05 |
| 1 | KLSF | | ✓ | | 28.32 | 47.79 | 62.83 | 22.43 | 55.62 | 71.42 | 48.06 | 69.20 | 86.56 | 91.66 | 82.47 |
| 2 | | | | ✓ | 29.86 | 50.66 | 64.15 | 23.27 | 56.90 | 73.14 | 49.66 | 69.28 | 86.85 | 91.81 | 82.65 |
| 3 | | ✓ | | | 28.98 | 49.33 | 61.94 | 24.73 | 55.88 | 72.87 | 48.96 | 66.14 | 84.92 | 91.04 | 80.70 |
| 4 | KLR | | ✓ | | 30.30 | 53.98 | 65.70 | 24.77 | 56.94 | 74.07 | 50.96 | 68.79 | 87.29 | 92.21 | 82.77 |
| 5 | | | | ✓ | 30.08 | 53.76 | 66.15 | 27.87 | 57.61 | 74.11 | 51.60 | 69.62 | 87.34 | 92.05 | 83.00 |

**Table 6: Comparisons between different key local feature selection methods in KLSF and comparisons between different mask methods in KLR on the RSITMD and CUHK-PEDES.**

retrieval. Similarly, Image KLR refines image local feature representations, which positively impacts unidirectional image retrieval. Consequently, a comparison between No.3 and No.1 reveals improvements in all R@K accuracy for image retrieval on the RSITMD dataset and in R@1 for image retrieval on the CUHK-PEDES dataset.

Text KLR and Image KLR are more focused on improving the unidirectional retrieval performance, and when used together, they improve the bidirectional retrieval performance. Comparing No.2 and No.1, significant increases in all R@K accuracy were observed on both RSITMD and CUHK-PEDES datasets, with the mR for bidirectional retrieval rising by 3.03% and 0.96%, respectively. The above experiments demonstrate the effectiveness of KLR.

(3) By comparing No.5 with No.4, we observe that the reconstruction with crossmodal information consistently outperforms the reconstruction with only intra-modal information, with an approximate 1% improvement in the mR on the RSITMD dataset. This indicates that crossmodal interaction benefits reconstruction tasks.

The above experiments demonstrate the effectiveness of each component in our EKLSR framework.

**Different Key Local Selection and Mask Strategies.** We investigate the selection method for key local features in the KLSF module and the masking methods in the KLR task in Table 6. The *random* represents a random selection (masking) method. The *topk* indicates selecting (masking) only the top k local features (segments) based on their importance factor ranking. The *our* represents the polynomial probability selection method, the characteristic of which is that segment features with higher importance factors have a greater probability of being selected, while also allowing segment features with lower importance factors to be selected.

Comparisons between No.2 and No.1 with No.0 indicate methods focusing on key local feature selection (*topk* and *our*) significantly outperform *random* selection on the mR metric. Similar trends

are observed in comparisons between No.5 and No.4 with No.3, with *topk* and *our* methods outperforming *random* mask. These highlight the pivotal role of key local features in specific domain tasks. Additionally, the *our* method performs better than the *topk* method in KLSF and KLR because it may also select (mask) image background regions and text words with low-importance factors. Although these segments have low importance factor, they are still essential for the overall understanding of the image and text.

## 5 CONCLUSION

To adapt the VLP model like CLIP to specific domain image-text retrieval tasks such as Text-Image Re-identification (TIReID) and Remote Sensing Image Text Retrieval (RSITR), we introduce the Explicit Key Local information Selection and Reconstruction (EKLSR) framework tailored for high similarity characteristics in specific domain data. Our Key Local information Selection and Fusion (KLSF) module leverages interpretable importance factors from CLIP's prior knowledge to identify key local features. These key token features, selected based on importance factors, are dynamically fused with instance-level global features to enhance feature representation in a shared space. Additionally, our approach employs Key Local segment Reconstruction based on multimodal interaction (KLR) to reconstruct key local segments of images (and text) using intra-modal contextual information and global information from matched modalities. This not only enriches the discriminative information of key local features but also intensifies the interaction between multimodal local features. Furthermore, our framework facilitates offline inference, catering to the real-time demands of specific domain applications. Ultimately, our EKLSR model attains substantial performance on two benchmark RSITR datasets and three TIReID datasets. In future work, we plan to conduct experiments on a broader range of VLP models.

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
