# OpenReview forum: "Selection and Reconstruction of Key Locals: A Novel Specific Domain Image-Text Retrieval Method"
_acmmm.org/ACMMM/2024/Conference — MM2024 Poster_

### Official Review · Reviewer_spzw · 2024-05-24

**Rating:** 4
**Confidence:** 3

**Summary:**

The paper presents an Explicit Key Local information Selection and Reconstruction Framework (EKLSR) to efficiently apply Vision-Language Pre-training (VLP) models in Specific Domain Image-Text Retrieval (SDITR) pursuits. The EKLSR overcomes issues of noise and uncertainty in current methods by selectively using key local information for feature representation. Demonstrated through experiments on five datasets, the EKLSR model outperforms existing models in Text-Image Person Re-identification and Remote Sensing Image-Text Retrieval scenarios.

**Strengths:**

1. The method proposed in this paper has been validated on RS and reID retrieval datasets, achieving good results. Additionally, ablation experiments have been conducted, proving that the method is effective.

2. The method is very intuitive.

**Limitations:**

1. The inference setting is not introduced in the paper. When performing retrieval on test datasets, does the author need to pre-process the entire dataset to get all importance factors for each token? And when selecting key local features, do you pick a fixed number based on importance factors (without randomness)?

2. The authors could explain more about why local features with lower similarity to the instance-level feature are more important. And the author could provide concrete examples to better illustrate what words are important and what are not.

3. The authors could conduct experiment to compare the proposed method to the baseline, i.e. CLIP, on the general domain. The proposed method seems not only works on specific domain, so it should be beneficial to the general domain.

4. In terms of writing, in 3.2 and 3.3, the authors would better decouple the intuitive and concise introductions from equations and formulas. And an error is found in figure 3: the color palette is with wrong color.

**Suitability:**

3

---

### Official Review · Reviewer_Zyut · 2024-05-25

**Rating:** 4
**Confidence:** 3

**Summary:**

The primary challenge of applying vision-language pre-training (VLP) models, such as CLIP, to Specific Domain Image-Text Retrieval (SDITR) lies in effectively leveraging discriminative fine-grained local information. Current approaches interact with all multimodal local features for alignment, which may introduce noise and uncertainty. To mitigate these issues, this paper proposes an Explicit Key Local information Selection and Reconstruction framework (EKLSR), which explicitly selects key local information to enhance feature representation. EKLSR introduces a Key Local information Selection and Fusion (KLSF) module and a Key Local segment Reconstruction (KLR) module based on multimodal interaction. Experiments conducted on five datasets across TIReID and RSITR validate the effectiveness of EKLSR. Notably, the EKLSR model achieves state-of-the-art performance on two RSITR datasets.

**Strengths:**

+ This paper proposes an Explicit Key Local information Selection and Reconstruction framework (EKLSR), which identifies key local features through Key Local information Selection and Fusion (KLSF) and Key Local segment Reconstruction (KLR), enriching the discriminative information of key local features.
+ The Key Local information Selection and Fusion part is simple and effective. Unlike MLM, KLR reconstructs key local information in images and text, which is also intriguing.
+ EKLSR achieves good performance on two specific domain image-text retrieval tasks.
+ The main text and supplementary materials present comprehensive experimental results.
+ The paper is written clearly and understandably.

**Limitations:**

+ In Section 4.2.2, how are the various hyperparameters ($\beta $, ...) determined?
+ The textual content and mathematical symbols in the figures must be consistent. For example, in Figure 2, is the CLS feature of the text right? What is the relationship between CLS and SOS/EOS?
+ Are all the parts that should be bolded in the table correctly bolded?
+ In the supplementary materials, the numbering of tables and figures should ideally follow the corresponding content in the main text.

**Suitability:**

3

---

### Official Review · Reviewer_Ymbw · 2024-05-25

**Rating:** 3
**Confidence:** 4

**Summary:**

Focusing on specific-domain image-text retrieval problem, this paper investigates on two main problems: 1) The interaction among local features leads to noise and uncertainty; 2) VLP models like CLIP are not specialized for local features, which perform unsatisfactorily. Therefore, “important factor” is proposed to measure the importance of local features, and a novel framework named Explicit Key Local Information Selection and Reconstruction is developed. In details, KLSF (Key Local information Selection and Fusion) selects key local features based on their importance factor, to avoid noise and uncertainty. KLR (Key Local segment Reconstruction) is proposed with Masked Image and Language Modeling with importance-factor-based masking for better local features.

**Strengths:**

This paper proposes a novel concept “importance factor”, and conducts extensive experiments on 5 datasets on Text-Image Person ReID and Remote Sensing Image-Text Retrieval.

**Limitations:**

A. The interpretability of “importance factor” is relatively low, where the visualization cannot support such too strong assumption.

According to Eq. 1 and 2, the closer to the global feature, the more important the local features are.

However, on one hand, if the background occupies the whole image like Figure 3, it is possible that there are some background information in the global features. As shown in the top-left and bottom-left picture in Figure 3, many no-object background parts are also highlighted with high “importance factor”.

On the other hand, if the foreground is the main part of image, like persons in Text-Image Person ReID task, it is also possible that some human parts are not comprised in global features. But no visualization of “importance factor” is provided for Text-Image Person ReID task like Figure 3. Similar uninterpretable highlighted parts can be also found in Figure 6 of supplementary materials.

Please provide more explanation about how such strong assumption is proven, as well as the visualization of “importance factor” for Text-Image Person ReID task.

B. The performance gain by the proposed method is marginal, which is conflict with the motivation.

In Figure 4 and Table 2-4, only about 3% performance gain are achieve in main-stream Text-Image Person ReID benckmarks. Moreover, each component merely obtain 1% performance gain in ablation study in Table 5 and 6 in Text-Image Person ReID task.

In Table 1, the proposed method brings 5% more improvements on RSITMD dataset for Remote Sensing Image-Text Retrieval, but still only 1% on RSICD dataset, compared with its baseline.

Given such marginal performance gain, it is confusing that whether the proposed method really solves the problem investigated in Figure 1. When placing the gained performance from Figure 4 into Figure 1(a), the results are still imbalanced with general domain Flickr30K.

Please explain such a conflict between the marginal performance gains and the motivation of balanced domain performances.

C. Some writing problems

1) In Row 2 and 3 of Table 5, KLSF is misspelled as “KTSF”.

2) In Eq. 12, masked features and global features are used to predict masked features. It should be non-masked features and global features for masked features.

**Suitability:**

3

---

### Official Review · Reviewer_7zPH · 2024-06-01

**Rating:** 5
**Confidence:** 3

**Summary:**

The paper proposes a novel framework named Explicit Key Local information Selection and Reconstruction (EKLSR) for Specific Domain Image-Text Retrieval (SDITR), particularly focusing on Text-Image Person Re-identification (TIReID) and Remote Sensing Image-Text Retrieval (RSITR). The framework enhances the discriminability of local features through two main components: Key Local information Selection and Fusion (KLSF) and Key Local segment Reconstruction (KLR). The authors demonstrate the effectiveness of their approach with experiments conducted on five datasets, achieving state-of-the-art performance on two RSITR datasets.

**Strengths:**

Novel Approach: The EKLSR framework addresses specific domain challenges by explicitly selecting and reconstructing key local features, which is a well-motivated and innovative solution.
Comprehensive Evaluation: The paper provides extensive experimental results on multiple datasets, showing significant improvements over existing methods.
Technical Depth: The integration of multimodal interaction-based reconstruction tasks (MLM and MVM) in fine-tuning is a novel application that enhances the discriminability of local features.
Clarity: The paper is well-written and clearly explains the proposed method, its components, and the underlying motivation.

**Limitations:**

Complexity: The proposed framework adds significant complexity to the baseline VLP models, which may impact the scalability and practical applicability in real-time systems.
Ablation Studies: While the ablation studies are thorough, additional insights into the interaction between different components could be beneficial. For instance, more detailed analysis on the impact of the polynomial probability selection method compared to simpler methods would add value.
Generalization: The paper primarily focuses on two specific domains (TIReID and RSITR). While the results are promising, the generalizability of the approach to other specific domains or broader applications remains uncertain.

**Suitability:**

3

---

### Meta-Review · Area_Chair_dojC · 2024-07-02

**Recommendation:** Accept (Poster)
**Confidence:** 4

**Metareview:**

There is a consensus that this paper is borderline, but three out of four are in favor of acceptance.
Two of the reviewers found that the authors' rebuttal provided relevant and satisfactory responses to the reviewers' comments in their rebuttal. One didn't discuss the rebuttal
There is one reviewer that is concerned about the explanation of Important Factor that is still not clear, I suggest to focus on this for the camera ready.
I decided to accept the rebuttal even if the style has been slightly modified, as mentioned by one reviewer.